# *Vitis vinifera* L. Leaves as a Source of Phenolic Compounds with Anti-Inflammatory and Antioxidant Potential

**DOI:** 10.3390/antiox14030279

**Published:** 2025-02-27

**Authors:** Nuria Acero, Jorge Manrique, Dolores Muñoz-Mingarro, Isabel Martínez Solís, Francisco Bosch

**Affiliations:** 1Departamento de CC Farmacéuticas y de la Salud, Facultad de Farmacia, Universidad San Pablo-CEU, CEU Universities, Urbanización Montepríncipe, 28660 Boadilla del Monte, Spain; jmanrique99@gmail.com; 2Departamento de Química y Bioquímica, Facultad de Farmacia, Universidad San Pablo-CEU, CEU Universities, Urbanización Montepríncipe, 28660 Boadilla del Monte, Spain; dmumin@ceu.es; 3Biomedical Sciences Institute, Universidad Cardenal Herrera-CEU, CEU Universities, Alfara del Patriarca, 46115 Valencia, Spain; isolis@uchceu.es (I.M.S.); fbosch@uspceu.es (F.B.); 4Departamento de Farmacia, Facultad de Ciencias de la Salud, Universidad Cardenal Herrera-CEU, CEU Universities, Alfara del Patriarca, 46115 Valencia, Spain; 5ICBiBE-Botanical Garden, University of Valencia, 46008 Valencia, Spain; 6Departamento de Ciencias Biomédicas, Facultad de Ciencias de la Salud, Universidad Cardenal Herrera-CEU, CEU Universities, Alfara del Patriarca, 46115 Valencia, Spain

**Keywords:** *Vitis vinifera*, phenolic compounds, anti-inflammatory, oxidative stress

## Abstract

*Vitis vinifera* is a plant known since ancient times mainly for the interest of its fruits. However, its leaves have traditionally been consumed as food in some regions of the Mediterranean basin and as a medicinal remedy. In this work, the phytochemical profile of this part of the plant, which is considered a bio-residue of viticultural processes, was analyzed (UHPLC-ESI(±)-QTOF-MS). Hydroxybenzoic acids, flavonols, and stilbenes are the main phenolic compounds identified. Its antioxidant and anti-inflammatory capacity were studied both in vitro and in cell culture. Grapevine leaves have a high capacity to scavenge free radicals, as well as to reduce oxidative stress induced by H_2_O_2_ in the HepG2 cell line. On the other hand, the methanolic extract of these leaves is capable of inhibiting lipoxygenase, an enzyme involved in inflammatory responses, with an IC_50_ of 1.63 μg/mL. In addition, the extract showed potent inhibition of NO production in LPS-stimulated RAW 264.7 cells. These results pointed out *V. vinifera* leaves as a powerful functional food with a high content of biologically active compounds. The enhancement of these by-products can be highly beneficial to food systems and contribute to the development of sustainable agriculture.

## 1. Introduction

*Vitis vinifera* L. is a deciduous climbing shrub that belongs to the Vitaceae Family. Its leaves turn red in autumn, and the fruits are edible berries of variable color: greenish, yellowish, reddish, purple, or blue [1]. Grapevine is a species native to the Mediterranean area, which has traditionally been consumed both as food and for its medicinal properties. The plant’s principal interest lies in its fruits, which are used to produce juices, wines, and liqueurs. These fruits are also consumed fresh or dried. Grapes and seeds are considered a source of pharmacological and cosmetic products. However, other parts of the plant, such as the leaves, have been less studied despite having been used for different therapeutic purposes. They have been traditionally recommended to stop bleeding inflammation, relieve pain, and diarrhea [2]. The fresh leaves are commonly applied topically to heal wounds and abscesses. Likewise, the leaves and other parts of the plant, such as the fruit and seeds, have been used from an ethnopharmacological perspective to prevent cardiovascular diseases, varicose veins, hemorrhoids, atherosclerosis, hypertension, inflammation due to injury or surgery, heart attacks, and strokes [3]. Currently, the European Medicines Agency accepts the well-established use of a dried aqueous grapevine leaf extract for the treatment of chronic venous insufficiencies symptoms. They are also consumed as food, including fresh, brined, or fermented [4]. The leaves of this plant are considered a waste product of agricultural production and the wine industry [5]. The study of the potential therapeutic applications of these leaves, as well as their use as a functional food, would mean a revaluation of this product in the framework of sustainable agricultural development [6].

In vitro and in vivo studies support *V. vinifera* traditional astringent, venotonic, and vaso-protective uses. The aqueous extracts of grapevine leaves have shown an antioxidant and protective activity of the vascular endothelium, increased capillaries resistance, and inhibited hyaluronidase, and thus achieving anti-inflammatory effects. Therefore, their use in patients with chronic venous insufficiency would be useful as it would increase oxygen supply and improve cutaneous microcirculation [7,8]. On the other hand, aqueous extracts from grapevine leaves have also demonstrated antidiabetic [9] and antibacterial and antifungal effects [10,11].

Several studies have been carried out with *V. vinifera* leaf extracts to demonstrate their possible protective role in skin cells against ultraviolet radiation. UV-A rays are responsible for the overproduction of reactive oxygen species (ROS) and the depletion of catalase, an antioxidant cellular enzyme. Grapevine polyphenols can protect keratinocytes against this oxidative stress, decreasing ROS and activating some antioxidant systems. In addition, ROS are involved in signaling pathways that induce pro-inflammatory cytokines, so these polyphenols have an anti-inflammatory potential of interest. The antioxidant effect shown by the extract could theoretically also protect against damage caused by these UV-B rays. However, results were not significant at the incubation times and doses tested [12].

Therefore, in view of the studies mentioned above, leaves from grapevine are postulated as an interesting source of nutraceuticals and therefore, of bioactive compounds of interest for human health [5].

The aim of this study is to analyze the effect of a methanolic extract of *V. vinifera* leaves against oxidative stress and inflammation by analyzing anti-inflammatory mechanisms of action other than those previously described. The extraction of *V. vinifera* leaves has been carried out at a low temperature and under a slightly acidic medium, which has the added value of preserving those compounds with low thermal stability and anthocyanins. Low-temperature extractions and those obtained under reflux can present important differences in their phytochemical profile because high temperatures promote the occurrence of the hydrolysis of many extract components [13,14]. The extract polyphenolic profile was analyzed using ultra-high performance liquid chromatography with electrospray ionization coupled to quadrupole-time-of-flight-mass spectrometry (UHPLC-ESI-QTOF-MS). The total content of phenols, flavonoids, and anthocyanins was also determined. The in vitro antioxidant activity of grapevine leaves was evaluated through their ability to scavenge several free radicals, which are involved in cellular oxidative stress, as well as other stable organic chemical free radicals. Subsequently, antioxidation and anti-inflammation assays were carried out in cultures of HepG2 and RAW 264.7 cell lines. The effect of the extract on intracellular ROS levels in the HepG2 line was studied, analyzing both its direct effect and its protective effect against induced oxidative stress. Finally, the in vitro extract effect on lipoxygenase and on the production of nitric oxide in the murine cell line RAW 264.7 previously stimulated with LPS (*E. coli* lipopolysaccharides) was assessed to evaluate its possible anti-inflammatory activity.

## 2. Materials and Methods

### 2.1. Chemicals and Reagents

Milli-Qplus185 system (Millipore, Billerica, MA, USA) was used as the source of purified water. LC–MS grade methanol was obtained from Honeywell. Chlorogenic acid (5-*O*-caffeoylquinic acid), cyanidin-3-*O*-glucoside, and quercetin-3-*O*-glucoside standards were supplied by Extrasynthesis Phytochemicals (Genay Cedex, France). MS grade formic acid (FA) was from Sigma-Aldrich Chemie GmbH (Steinheim, Germany). The rest of the chemicals, including enzymes, substrates, free radicals, cell culture mediums, and supplements, were purchased from Sigma-Aldrich (St. Louis, MO, USA).

### 2.2. Plant Material and Extraction Procedure

The leaves were provided by Plameca (Suanfarma S.A., Barcelona, Spain) (batch n° 300741), a food supplements and medicinal plants laboratory and supplier. This implies that they come from organic crops where no phytosanitary products have been used. Leaves were collected in Bulgaria in September of 2019. The *Vitis vinifera* variety was verified through DNA analysis using microsatellite markers conducted by the Institute of Vine and Wine Sciences (ICVV) (University of La Rioja, La Rioja, Spain). The comparison of the obtained DNA profile with the genotypes included in the ICVV databases and the International Catalogue of Grape Varieties (VIVC, www.vivc.de, accessed on 4 September 2024) revealed a single complete match with Merlot (VIVC 7657).

To obtain the extract, 20 g of pulverized *V. vinifera* mature leaves were extracted with 200 mL of 80% methanol with 0.1% commercial HCl, which helped to preserve the anthocyanins [15,16]. The mixture was introduced into an ultrasonic bath (Ultrasons, Selecta^®^, Barcelona, Spain) for 15 min. It was then kept in stirring for 30 min and sonicated again for another 15 min. Subsequently, the sample was filtered, and 200 mL more 80% methanol with 0.1% commercial HCl were added to grapevine leaves marc (left plant material after the extraction process), repeating the previous stirring and sonication steps. The obtained methanolic extract was evaporated in a rotavapor (Buchi, R-200, Flawil, Switzerland) at a temperature below 40 °C. The yield of the extract was 15.2% (*w*/*w*).

### 2.3. Phytochemical Analysis

#### 2.3.1. Total Phenolic Content

The extract’s total phenolic content was determined by the Folin-Ciocalteau colorimetric method [16]. A total of 5 μL of five different concentrations of the extract were added to each well of a 96 plate and mixed with 80 μL of 10% Folin-Ciocalteau. Five min later, 160 μL of 7.5% sodium carbonate (Na_2_CO_3_) was added, and the plate was shaken. After 30 min of incubation in darkness, the plate was read on a Spectrostar-Nano reader (BMG Labtech, Ortenberg, Germany) at 765 nm. Gallic acid was used as standard (500–29.26 µg/mL). Results were expressed as mg of gallic acid equivalents per 100 mg of extract. All measurements were carried out in triplicate.

#### 2.3.2. Total Flavonoid Content

A colorimetric assay was performed as described below to determine the concentration of total flavonoids present in the extract [15]. Epicatechin was used as standard (0.1–0.013 µg/mL). A total of 200 μL of different concentrations of the extract, 200 μL of Mili-Q water as blank, or 200 μL of different concentrations of epicatechin, 800 μL of Mili-Q water, and 60 μL of 5% NaNO_2_ were mixed. After five minutes, 60 μL of 10% AlCl_3_ was added, and then, the next minute, 400 μL of 1 M NaOH was added. Color appeared immediately, and the absorbances against the blank were read at 510 nm in the Spectrostar-Nano spectrophotometer (BMG Labtech, Ortenberg, Germany). Results were expressed as mg of epicatechin equivalents per 100 mg of extract. All measurements were conducted in triplicate.

#### 2.3.3. Total Anthocyanin Content

This determination was made using the differential pH method [17]. Two buffer solutions were prepared: 0.2 M KCl pH 1 and 1 M sodium acetate pH 4.5. A total of 7000 μL of buffer was added to 500 μL of extract (1 mg/mL), obtaining two mixtures, one with each of the buffers. After 15 min incubation at room temperature protected from light, absorbances of both solutions were measured at 510 and 700 nm. All measurements were carried out in triplicate. Absorbance was calculated using the formula:Abs = [(Abs510 − Abs700)pH 1.0 − (Abs510 − Abs700)pH 4.5].

The anthocyanin concentration was calculated using the equation:C (mg/L) = (Abs/ε × L) × Mw × dilution factor × 10^3^,
where: ε is cyanidin-3-glucoside molar extinction coefficient of = 26,900 L/mol·cm; Mw is cyanidin-3-glucoside molecular weight = 449.2 g/mol; and L is the cuvette optical path length = 1 cm.

#### 2.3.4. Polyphenolic Profile Determination by UHPLC-ESI-QTOF_MS Analysis

The analysis of the samples was carried out using a 1290 Infinity series UHPLC system, which was equipped with an electrospray ionization (ESI) source featuring Jet Stream technology. This system was coupled to a 6545 iFunnel QTOF/MS (Agilent Technologies, Waldbronn, Germany) [17].

Chromatographic separation was achieved on a Zorbax Eclipse XDB-C18 reverse-phase column (4.6 × 50 mm, 1.8 µm) (Agilent Technologies, Santa Clara, CA, USA), maintained at 40 °C. The mobile phase consisted of solvent A: 0.1% formic acid in water, and solvent B: methanol, with a flow rate set at 0.5 mL/min. The gradient elution program was as follows: 2% B for 0–6 min, 2–50% B for 6–10 min, 50–95% B for 11–18 min, and 95% B for 2 min (18–20 min). The gradient was returned to 2% B within one minute (20–21 min) for re-equilibration, resulting in a total analysis time of 25 min. The injection volume was 2 µL. The mass spectrometer operated in full scan mode (*m*/*z* 50 to 1500) with a scan rate of 1 scan/s in both positive and negative ESI modes. MS/MS data were acquired automatically in Auto MS-MS scan mode, utilizing 30 eV of collision energy.

Accurate mass measurement was maintained via an automated calibration system, which continuously introduced reference solutions containing *m*/*z* 121.0509 (protonated purine) and *m*/*z* 922.0098 (protonated HP-921) for positive ESI mode, and *m*/*z* 119.0363 (deprotonated purine) and *m*/*z* 966.0007 (formate adduct of HP-921) for negative ESI mode. The capillary voltage was set to ±4000 V for both ionization modes. The source temperature was maintained at 225 °C, with nebulizer and gas flow rates of 35 psig and 11 L/min, respectively. The octopole RF voltage (OCT RF Vpp) was set at 750 V, and the fragmentor voltage was 75 V.

Data acquisition and system control were managed using MassHunter Workstation Software LC/MS Data Acquisition Version B.07.00 (Agilent Technologies, Santa Clara, CA, USA). Data processing was performed in MassHunter Qualitative Analysis Software Version B.08.00 (Agilent Technologies, Santa Clara, CA, USA).

### 2.4. In Vitro Antioxidant Assays

#### 2.4.1. DPPH^•^ Radical Scavenging Assay

The ability of the extract to reduce DPPH^•^ radical by donating a proton was determined colorimetrically [18]. A total of 100 µL of DPPH (1 mM) was mixed with 100 µL of methanol (blank), sample, or ascorbic acid (standard) at different concentrations. Absorbance was measured at 517 nm after 20 min of incubation and preserved from light at room temperature in a plate reader (Spectrostar-Nano BMD Labtech, Ortenberg, Germany). The IC_50_ (extract and standard concentration, which reduces 50% of the radical) was calculated.

#### 2.4.2. ABTS Radical Scavenging Assay

The ABTS radical (2,2′-azinobis-(3-ethylbenzothiazoline-6-sulfonic acid) scavenging assay was performed following the Acero et al. protocol [15] with slight modifications. Reagents were all prepared in PBS, and the assay was performed on a 48-well plate. In each well, 400 µL of ABTS (500 µM), 12 µL of myoglobin III (myoglobin 400 µM: K4Fe(CN)6 740 µM, 1:1), 20 µL of PBS, extract, or standard (ascorbic acid) at different concentrations, and 228 µL of water were mixed. Just before the plate reading, 340 µL H_2_O_2_ (450 M) was added. Absorbance was measured on a Spectrostar-Nano plate reader (BMG Labtech, Ortenberg, Germany) every minute for 10 min, after 6 min of incubation. Extract and standard IC_50_ were calculated.

#### 2.4.3. Xanthine/Xanthine Oxidase Assay

The xanthine–xanthine oxidase enzyme catalyzes the oxidation of hypoxanthine to xanthine and then to uric acid. In this process, superoxide radicals are generated. The ability of the extract to inhibit the enzyme or/and to scavenge superoxide radicals (O_2_^•−^) was analyzed [15]. Reagents were prepared in phosphate buffer (50 mM KH_2_PO_4_/KOH, pH 7.5). In a 96-well plate, 62.5 µL of phosphate buffer, 10 µL of 15 mM EDTA (ethylenediaminetetraacetic acid), 15 µL of 3 mM hypoxanthine, 25 µL of 0.6 mM nitroblue tetrazolium, 25 μL of phosphate buffer as blank, or different concentrations of the extract, or standard (ascorbic acid), and 25 μL of xanthine oxidase (1 U/10 mL) were added to each well. Measurements were performed at 37 °C each 5 min for a total time of 40 min after the enzyme addition on a Spectrostar-Nano (BMG Labtech, Ortenberg, Germany) plate reader at 560 nm. Based on the results, IC_50_ was calculated.

The capability of the extract to inhibit the xanthine oxidase enzyme was estimated by measuring the uric acid generation since it corresponds to the final product of the reaction. 87.5 μL of phosphate buffer, 10 μL of EDTA (15 mM), 15 μL of 0.1 mM xanthine, 12.5 μL of phosphate buffer, or different concentrations of the extract or standard (ascorbic acid), and 25 μL of xanthine oxidase (1 U/10 mL) were mixed. Measurements were performed at intervals of 5 min for 40 min at 295 nm. Results were expressed as IC_50_ of the enzyme activity.

#### 2.4.4. Hydroxyl Scavenging Assay

For this assay, Acero et al. protocol was followed [15]. 400 µL of 1.5 mM FeSO_4_, 280 µL of 6 mM of H_2_O_2_, and 200 µL of water, extract, or standard at different concentrations were mixed in the presence or absence of 120 µL of 20 mM sodium salicylate. After 1 h of incubation at 37 °C, absorbance was measured at 562 nm (Spectrostar-Nano plate reader, BMG Labtech, Ortenberg, Germany). The hydroxyl radical scavenging percentage and the IC_50_ were calculated.

#### 2.4.5. Nitrogen-Free Radical Scavenging Assay

Griess reaction was used to estimate nitric acid scavenging activity [15]. A total of 200 µL of 10 mM sodium nitroprusside, 50 µL of PBS, and 50 µL of the methanolic grapevine leaf extract, or standard (ascorbic acid) dissolved in methanol at different concentrations, were incubated for 150 min at 25 °C preserved from light. Then, 50 µL of the mixture and 100 µL of sulphanilic acid reagent (0.33% of sulphanilic acid in 20% of glacial acetic acid) were mixed. Five min later, 100 μL of 0.1% N-(1-Naphthyl)-ethylenediamine dihydrochloride was added. Absorbance was measured after 30 min at 25 °C at 540 nm in a Spectrostar-Nano (BGM Labtech, Ortenberg, Germany) plate reader. IC_50_ was calculated based on the results.

Three independent experiments were conducted in triplicate for all antioxidant in vitro tests.

### 2.5. Cell Culture

Two cell lines were used in the present study: murine macrophages, RAW 264.7 (Ref ECACC 91062702), and human hepatocarcinoma, HepG2 (Ref ECACC 85011430). In both cases, cells were maintained in Eagle’s minimum essential medium (EMEM) supplemented with Glutamine 2 mM, 1% of non-essential amino acids, 10% of fetal bovine serum (FBS), and 1% of antibiotics (10,000 U of penicillin and 10 mg of streptomycin/mL). Cells were incubated at 37 °C under a 5% CO_2_ atmosphere.

The FBS percentage with which the medium was supplemented was reduced to 1% during cell assays to prevent the phenolic compounds of the extract from interfering with serum components, which could lead to artifacts with cytotoxic activity [19].

### 2.6. Vitis vinifera Extract Cytotoxicity

Cells were seeded in 96-well plates at a rate of 10,000 cells/well in a culture medium supplemented with 1% FBS. After 24 h of incubation, 50 μL of the different concentrations of the extract dissolved in 1% FBS medium were added, using 1% FBS medium as a control. A negative cell-free control was also performed. The plates were incubated for 72 h at 37 °C. After this time, the medium was aspirated, and 150 μL of medium and 50 μL of MTT (3-(4,5-dimethylthiazone-2yl)-2,5-diphenyltetrazole bromide) 0.1% in PBS were added. After 4 h of incubation, the content of the well was aspirated, and 100 μL of DMSO was added to dissolve formazan crystals. Absorbance was measured on a Spectrostar-Nano (BGM Labtech, Ortenberg, Germany) plate reader at 550 nm [20]. Results were expressed in percentages of cell growth inhibition.

### 2.7. Antioxidant Cell Culture Assay

The effect of *V. vinifera* leaf extract on HepG2 cell’s intracellular ROS levels was analyzed. Two culture situations have been studied: the direct effect, which evaluates the impact of different extract concentrations on ROS levels of cells growing under normal culture conditions, and its protective effect, in which cells were pretreated with the extract before oxidative stress was induced.

To determine intracellular ROS levels, the diacetate 2-7dichlorodihydrofluorescein (DCFH-DA) assay was performed on the HepG2 cell line [21]. Fifteen thousand cells per well were seeded in 96-well plates. They were incubated for 24 h at 37 °C and an atmosphere of 5% CO_2_, in the corresponding medium supplemented with 1% FBS. After 24 h of incubation, the direct or protective effect was determined as detailed below.

To measure the direct effect, after the 24 h incubation referred to above, the medium was aspirated, and 0.02 mM of the DCFH-DA reagent was added. After 30 min, well content was aspirated, cells were washed with PBS, and different concentrations of grapevine extract diluted in culture medium were added.

To determine the protective effect after 24 h of incubation, the medium was aspirated, and different concentrations of *V. vinifera* extract diluted in culture medium were added. The cells were pre-treated for 24 h. After this time, the medium was aspirated, DCFH-DA 0.02 mM was added, and incubated for 30 min. Then, the dye solution was removed, cells were washed with PBS, and 200 mM H_2_O_2_ diluted in the culture medium was added.

Fluorescence was measured from the time the extracts were added in the direct effect assay, or when H_2_O_2_ was added in the protective effect measurement, every 15 min, for a total of 90 min in Fluorstar optima plate reader (BMG Labtech, Ortenberg, Germany), at an excitation wavelength of 485 nm, and 529 nm of emission. The DCFH oxidation by intracellular ROS leads to 2′-7′-dichlorofluorescin which is fluorescent. Therefore, fluorescence was proportional to the sample ROS levels. Results were expressed as % fluorescence with respect to the control.

### 2.8. Anti-Inflammatory Activity

#### 2.8.1. Inhibition of Nitric Oxide Production by RAW 264.7

The inhibition of nitric oxide (NO) production evaluated in RAW 264.7 cells previously activated with LPS, following the Griess reaction method was used to assess the anti-inflammatory potential of *V. vinifera* methanolic leaf extract [22].

RAW 264.7 cells were plated in a 96-well plate at a density of 6 × 10^5^ cells per well and incubated 24 h at 37 °C in a 5% CO_2_ atmosphere. After incubation, the cells were treated with either culture medium (negative control), 1 μg/mL of LPS (positive control), or with LPS and varying concentrations of grapevine leaf extract (37.5, 75, 125, 250, and 500 μg/mL) for an additional 24 h. Subsequently, 100 μL of culture medium from each well was transferred, and 90 μL of 1% sulphanilamide in 5% phosphoric acid solution was added. Following a 5 min incubation at room temperature, 90 μL of 0.1% N-(1-naphthyl)-ethylenediamine was added. After 30 min in the dark absorbance was measured at 550 nm using a plate reader (Spectrostar-Nano, BMD Labtech, Ortenberg, Germany). Results are presented as the percentage of NO production relative to the control.

#### 2.8.2. Lipoxygenase Inhibition Assay

The lipoxygenase inhibition assay was carried out according to the protocol by Matos et al. [23].

A total of 500 µL of phosphate buffer (100 mM, pH 8), 100 µL of different concentrations of the extract in buffer, and 200 µL of soybean lipoxygenase 1-B (E.C. 1.13.11.12, Sigma-Aldrich, Burlington, MA, USA) were mixed to achieve 167 U of enzyme in the final volume. After 10 min at 25 °C, 200 µL of 0.67 mM linoleic acid was added to initiate the reaction. Absorbance was measured at 234 nm every 30 s for 4 min. The percentage of enzyme inhibition was calculated. Quercetin was used as a positive control.

### 2.9. Statistical Analysis

Three independent experiments were conducted in triplicate for all in vitro tests.

Three independent experiments were conducted for cell culture assays, with 8 repetitions per analysis.

After the homogeneity of variance was verified using the Levene test, the statistical analysis was carried out through one-factor ANOVA followed by Bonferroni test post hoc comparison (*p* ≤ 0.05). The IBM SPSS Statistics 24 program was used.

## 3. Results and Discussion

### 3.1. Phytochemical Analysis

*V. vinífera* leaves have interest both to be used as food and as herbal medicine [24]. It is known that they are rich in phenolic compounds, such as flavonoids or tannins [25]. Another characteristic of these leaves is that they acquire a red color during autumn, which points to high anthocyanin concentrations during this period [26]. The leaves’ chemical composition not only changes through the plant’s vegetative cycle but also depending on the weather or the soil in which the plant grows [27,28]. The harvesting of vine leaves for phytotherapeutic purposes is usually done when the leaves are still green, usually just after the grapes have been harvested. This is the case of the leaves used in this study that were collected before they changed to red autumn color (September 2019). The anthocyanin concentration is therefore still very low at this sampling moment, and it was not detectable by the spectrophotometric method. Other studies carried out on grapevine green leaves from northern Serbia, Brazil or Poland confirm the absence of significant amounts of this type of compound [29,30]. Total phenol and flavonoid content results are shown in Table 1 as a percentage of gallic acid, or epicatechin equivalents in the extract, respectively.

As can be shown, the extract has a high content of 25.03% of total phenols and 7.05% of total flavonoids. Other white and red grapevine varieties from Portugal showed values between 11.2 and 15% of total phenols [31]. On the other hand, Dani et al. [29] observed percentages of 20.2 and 19% in leaves of this plant organic crops in Brazil. The results confirm the high content of this type of secondary metabolites in our sample and highlight the influence of the extraction method, as well as the variety, weather, and soil on the concentration of active ingredients in plants [13,32]. The flavonoid content of the leaves analyzed by Lima et al. varied between 3.9 and 5.4% measured as a percentage of quercetin equivalents, so a direct comparison cannot be made with our results [31]. Similarly, other studies showed even lower flavonoid amounts in Brazilian plant leaves, ranging between 7.94 and 8.95 μg per 100 mg of extract, but also measured using different analytical assays. However, our results point, as in the case of total phenols, to high percentages of these compounds directly involved in the antioxidant activity of plant extracts [33].

The UHPLC-ESI-QTOF-MS analysis provided us with an accurate method for rapid and efficient characterization of the major phenolic compounds from *V. vinifera* leaves methanolic extract. Table 2 lists those compounds that have been identified with pure standards according to their retention time, accurate mass, isotopic profile, and MS/MS spectral data.

Most of these compounds have been described previously [27,29,30,34,35] such as quercetin and kaempferol glucosides, or caffeic acid, both in *V. vinifera* and also in other species leaves such as *V. labrusca* var. Bordo. On the other hand, Goufo et al. carried out a detailed study publishing a reference list of phenolic compounds present in different parts of the grapevine plant, including the leaves [25]. To the best of our knowledge, morin, and phloretin have been identified for the first time in the leaf of *V. vinifera*. The presence of shikimic acid, although it is not a phenol, has been included in the table as a precursor for the synthesis of many of the phenolic compounds.

The UHPLC-ESI(−)-QTOF-MS analysis allowed us to obtain the experimental accurate masses for 8 stilbenes derivatives. This led, together with their retention times (tR (min)) and the in-source fragmentation observed, to the tentative identification of these significant compounds, for which no standards were available. Data were compared against online databases such as FOODB (https://foodb.ca/, accessed on 6 March 2024) and by using the tool developed by our University CEU mass mediator (http://ceumass.eps.uspceu.es/, accessed on 6 March 2024).

The *m*/*z* experimental values from the Base Peak Chromatograms for compounds 1-8, obtained in negative ion mode, were compared with those reported in the databases and literature and are summarized in Table 3 [25].

Following acquired MS data, stilbene aglycones, glucoside, and glucuronide derivatives, as well as dimers, were traced in the studied methanolic extract (Table 3). Under the conditions used, most of the detected compounds had intensive signals corresponding to the deprotonated adduct [M-H]^−^ (Figure 1). MS data cannot allow for distinguishing between viniferin, resveratrol, or pterostilbene isomers. However, resveratrol isomers were discerned according to relative abundance (*trans*-resveratrol isomer) which is more stable and abundant [36] and based on comparison with the literature data [25].

### 3.2. In Vitro Scavenging Capacity of Vitis vinifera Leaves

In view of the high phenol and total flavonoid content of the extract, its antioxidant effect was analyzed, starting with its ability to scavenge free radicals in vitro. Chemical radicals such as DPPH and ABTS were tested, as well as other radicals largely responsible for cellular oxidative stress, such as O2^•−^, OH^•^ or NO^•^. The methanolic extract of grapevine leaves contains a broad range of bioactive compounds, which means that evaluating its antioxidant activity requires multiple assays. Each method varies in terms of the sensitivity of the procedure, how radicals are generated, and the approach used to measure the inhibition reaction endpoint. Therefore, it is essential to use more than one assay and reference substances, for example, ascorbic acid is used concretely in this study (except for NO• assay where gallic acid was used), to obtain useful information on the antioxidant capacity of a complex phytoextract [37]. The results, expressed as IC_50_ values, are shown in Table 4.

The ability of the extract to scavenge DPPH and ABTS radicals is noticeably lower than that of the reference substance, being ascorbic acid 10 times more active for the DPPH assay, and twice for the ABTS assay. However, interesting results were obtained for those tests that studied the ability of *V. vinifera* leaves to scavenge radicals involved in cellular oxidative stress processes such as superoxide radicals, hydroxyl, or N free radicals.

Xanthine oxidase is an enzyme that catalyzes the oxidation of hypoxanthine and xanthine to uric acid, producing in the process reactive oxygen species. The overproduction of uric acid by xanthine oxidase is associated with gout and other diseases such as diabetes, cardiovascular disorders, or nephrolithiasis [38]. In addition, this enzyme increases oxidative stress due to the excessive production of superoxide radicals, which can start pathologies such as inflammation, atherosclerosis, cancer or simply aging. Therefore, advances in the prevention of diseases caused by oxidative stress can be approached from this enzyme as a possible therapeutic target [39,40]. In this assay, inhibition in the reduction of NBT is determined, which indicates a decrease in the amount of ROS, a possible consequence of both the inhibition of the enzyme (decrease in the production of uric acid), and/or an uptake of the radical by the active ingredients of the extract, or reference substance. At the tested concentrations, neither *V. vinifera* leaf extract nor ascorbic acid inhibited xanthine oxidase. However, they show the ability to capture the superoxide radical with a very similar IC_50_ (Table 4).

Oxidative stress alters cellular homeostasis, initiating numerous degenerative processes in cells. The superoxide radical (O_2_^−^) can be transformed to hydroxyl radical in certain processes. This radical (hydroxyl) is highly harmful, as it initiates chain reactions that damage genetic material, proteins, and other biomolecules that are essential for cellular life. Therefore, it triggers processes such as aging, mutagenesis, or carcinogenesis [40]. The removal of hydrogen peroxide is also vital to prevent the formation of OH^•^ radicals that destabilize the redox balance, which has, as already mentioned, a direct relationship with the onset and progression of various pathologies [41,42]. The extract’s ability to capture the OH^•^ radical is markedly superior to that of the reference substance, with an IC_50_ of 155.77 and 279.53 μg/mL, respectively.

Endothelial vascular dysfunction is a determinant factor of cardiovascular disease in the elderly, due to an excess of superoxide radical and hydrogen peroxide that limit the vasodilator function of nitric oxide [43]. The preservation of the functionality of the vessels is decisive in aging to avoid processes related to oxidative stress and inflammation. Therefore, the beneficial effects of grapevine leaf extract could act on vascular deterioration and prevent these pathologies. In this sense, NO has different functions, highlighting its role as a vasorelaxant [44]. Some diseases such as diabetes, inflammatory diseases, septic shock, or adult respiratory distress syndrome, among others, have been associated with NO overproduction. In this sense, NO^•^ scavengers have demonstrated pharmacological interest [45]. Grapevine leaf extract scavenge NO^•^ radical more efficiently, approximately twice than the reference substance, which in this case was gallic acid since ascorbic acid was not able to capture this reactive species of N at tested concentrations.

Some stilbenes of the extract such as resveratrol and viniferin, have shown NO^•^, hydroxyl, and superoxide radical scavenging capacity. In addition, these molecules are able to interact with other polyphenols, for example, stilbenes with each other in an additive way, or resveratrol with caffeic acid with synergistic effects, increasing the antioxidant capacity of complex extracts [46,47]. Several studies have been carried out on the antagonistic, additive, or synergistic interactions between plant polyphenols [48]. The antagonistic interactions have mainly been shown with molecules that belong to the flavonoid and phenolic acid families, such as catechin, quercetin, caffeic acid, kaempferol, and gallic acid [46,47]. These interactions among phenolics may promote changes in overall antioxidant capacity, which is difficult to predict on the basis of their individual antioxidant capacities. Although grapevine studies focus on resveratrol due to its high antioxidant capacity, other components of the sample, mainly other phenolic compounds, are of great importance due to stated reasons [49]. In this sense, the way in which the extract has been carried out in this study has made it possible to preserve many compounds that may affect the antioxidant capacity of the mixture. The free radical scavenging activity of the studied *V. vinifera* leaves extract points out it as a rich source of phenolic compounds, which can be used as a natural antioxidant product or an additive for different purposes.

Other byproducts of viticulture production, such as grape pomace, have demonstrated antioxidant capacity by scavenging superoxide, ABTS, and NO radicals with IC_50_ values of 74.17, 7.79, and 15.34 µg/mL, respectively. These results led Tapia et al. [50] to conclude that grape pomace has potential applications in pharmacology, cosmetics, and the food industry. The results obtained with the methanolic extract of *V. vinifera* leaves show a greater or similar capacity to scavenge these radicals compared to grape pomace, which justifies the need for further studies to validate their practical utilization.

Grapes contain high levels of phenolic compounds, which confer significant antioxidant activity, and, consequently, health-promoting properties [51]. A study reveals that, among a range of popular beverages, both grape juice and red wine are among the top four with the highest in vitro antioxidant potential of all the beverages evaluated, surpassing orange juice, iced tea, and apple juice, although falling behind pomegranate juice [52]. Grapes exhibit IC_50_ values for DPPH (270 ± 1 µg/mL) and ABTS (40 ± 3 µg/mL) that are notably higher than those reported in this study, suggesting that the leaves possess superior antioxidant capacity. Furthermore, while the skin and seeds of these fruits are recognized as antioxidant-rich sources with potential for use as dietary supplements, their antioxidant capacity is lower than that of the leaves in these assays, with the exception of the seeds in DPPH capture (IC_50_: 30.6 ± 3.4 µg/mL [53].

### 3.3. Cytotoxicity of Vitis vinifera Leaves Extract on HepG2 and RAW 264.7 Cell Lines

Methanolic extract from leaves of *V. vinifera* showed no cytotoxic effect on Hep-G2 and RAW 264.7 cell lines at doses between 500 and 50 μg/mL. No significant differences (ANOVA, Bonferroni *p* < 0.05) were detected between the growth percentage of the control cells and those treated for 72 h with the highest tested concentration, although a small not significant decline in growth was detected (around 90% of growth in both cell lines at the highest tested concentration).

### 3.4. Cell Culture Radical Scavenging Activity of Vitis vinifera Leaves

Once the absence of cytotoxicity was confirmed, the effect of grapevine leaf extract at different concentrations (from 500 μg/mL to 66.74 μg/mL through ¾ serial dilutions) on intracellular ROS levels in human hepatocellular carcinoma cell culture (HepG2 line) was studied. Both the direct effect of the extract on the cells and the pre-treating protective effect against induced oxidative stress (H_2_O_2_) were analyzed. Results are shown in Figure 2 and Figure 3.

As shown in Figure 2, direct treatment with grapevine leaf extract produces a dose-dependent increase in intracellular ROS levels. As the concentration of extract increases, ROS levels rise, probably due to the presence of flavonoids, compounds that, depending on their concentration, could act as anti- or pro-oxidants [54]. The pro-oxidant activity has also been detected with flavonoids and with the flavonoid fraction of other botanical genera [55,56]. This ability to increase ROS causes the proposal of maximum doses to avoid an unwanted event due to the increase in ROS. Even so, a moderate pro-oxidant effect may be advantageous, as it can provide improved protection against oxidative stress, through the stimulation of the cell’s antioxidant defenses. The protective effect of grapevine leaf extract was further investigated to determine whether the observed rise in intracellular ROS levels could be classified as mild or moderate, and thus beneficial for the cell. To determine whether the observed rise in intracellular ROS levels could be classified as mild or moderate, and thus beneficial for the cell, the protective effect of grapevine leaf extract was further investigated.

Cells were pretreated with different concentrations of the extract for 24 h, and then oxidative stress was induced with H_2_O_2_. The results are shown in Figure 3. It can be observed that, after 90 min of stress induction, there are significant differences between the control of unstressed cells and that of stressed cells. Treatment with grapevine extract significantly reduces intracellular ROS levels at all tested concentrations, with a noticeable dose–response effect.

Aqueous extracts from leaves of other species of *Vitis*, such as *V. lambrusca*, have demonstrated their ability to reduce lipid and protein damage induced by hydrogen peroxide in rat brains. The leaves of this species, and specifically its phenolic compounds, were able to reverse the effect of H_2_O_2_ on antioxidant enzymes such as superoxide dismutase or catalase [29]. Consequently, Dani et al. [29] conclude that *V. lambrusca* leaves could be used to delay or prevent the development of diseases associated with oxidative stress, such as neurodegenerative diseases. These considerations could be extrapolated to the leaves of *V. vinifera*, so new studies related to the antioxidant mechanism of the extract would be justified.

### 3.5. Anti-Inflammatory Capacity Assays

#### 3.5.1. Measurement of NO Production in RAW 264.7 Cells

Nitric oxide (NO) is involved in neuronal communication, vasodilation, and neurotoxicity processes, but it is also an inflammatory marker. Overproduction of NO induces tissue damage associated with inflammation. NO is produced by macrophages, which allow leukocyte migration and immediate defense against foreign agents. Components of the Gram-negative bacteria cell walls, such as lipopolysaccharides (LPS) are capable of activating macrophages. LPS stimulates the production of pro-inflammatory mediators such as prostaglandins and NO, enzymes like cyclooxygenase-2 (COX-2) and inducible nitric oxide synthase (iNOS), as well as pro-inflammatory cytokines including IL-6, IL-1β, and TNF-α [57].

The change in NO production was measured in LPS-stimulated RAW 264.7 cells to assess the anti-inflammatory potential of grapevine leaf extract. The results, presented in Figure 4, are shown as the percentage of NO production relative to the positive control.

A dose-dependent reduction in NO production was observed in cells treated with the grapevine leaf extract. As the concentration of the extract increased, NO production decreased. Even at lower doses, grapevine extracts led to a significant reduction in NO production.

Plants are thought to be an important source of natural antioxidants, mainly due to their content in phenolic compounds. In addition, these kinds of substances have shown potent inhibiting NO production activity [58]. Consequently, there is a growing interest in the search for antioxidants from plants to inhibit the production of this important signaling molecule [59]. Some dietary sources, such as fruits and vegetables prevent diseases due to their high content of polyphenols with free radical scavenging capacity [60,61]. As mentioned in the introduction, there are numerous phenolic compounds in the hydroalcoholic extract of *V. vinifera* leaf, such as, among others, flavonoids, catechins, or resveratrol that have anti-inflammatory activity. Aouey et al. reported grapevine leaves’ extract ability to decrease vascular permeability in a dose-dependent manner in paw edema in mice supports the traditional use of the plant [35]. Our results suggest that the reduction of NO levels is also responsible for the anti-inflammatory activity of those components of the extract. The ability to reduce LPS-inducible inflammatory mediators is regarded as a useful indicator of the anti-inflammatory activity of plant extracts [62]. Resveratrol is known to possess anti-inflammatory activity, through several mechanisms, including the downregulation of iNOS expression, that leads to a reduction in NO production as well as IL-6 in RAW 264.7 cells [63]. Other phenols present in the studied *V. vinifera* leaves extract, such as flavonoids like apigenin, kaempferol, luteolin, or quercetin also inhibit NO production through iNOS enzyme expression [58,64].

On the other hand, it has been observed that the aqueous extract of *V. vinifera* leaves can inhibit in vitro the secretion and expression of IL-8 induced by TNF-α, counteracting inflammation in the cells of the gastrointestinal epithelium and improving the state of health [65]. In addition, leaf phytochemicals showed, synergistically, a reduction in pro-inflammatory mediators released by keratinocytes in chronic skin pathologies such as psoriasis [66]. These results reinforce the possibility of using these plant leaf extracts for the development of food supplements and cosmetic or pharmaceutical products.

#### 3.5.2. Inhibition of Lipoxygenase (LOX)

Lipoxygenases are pro-inflammatory enzymes that catalyze the oxidation of polyunsaturated fatty acids to give hydroperoxides. These enzymes are widespread in humans and play an important role in inflammatory reactions. The release of pro-inflammatory cytokines, which activate lipoxygenases, can be triggered in response to high levels of ROS. These enzymes start leukotrienes and prostaglandins synthesis that are related to several diseases. Consequently, lipoxygenase inhibition is considered an interesting mechanism for disease prevention. In this sense, there is an increasing interest in plant-derived remedies [67].

The effect of *V. vinifera* leaves extract and quercetin, which was used as a reference substance, on lipoxygenase is shown in Figure 5. From the obtained data, the IC_50_ was calculated, which was 1.63 µg/mL for the grapevine leaf and 1.57 µg/mL for quercetin.

As can be seen in Figure 5, the effect of the extract and that of the reference substance are very similar. The presence of quercetin and several of its heterosides could, at least in part, justify these results. However, other compounds could be acting additively or synergistically [68], although new experiments are necessary to support this hypothesis. Further components of the extract such as stilbenes, should be also responsible for this capability. In this sense, resveratrol and viniferin also have shown potent activity against lipoxygenase, with IC_50_ values in the low μM range [69,70,71,72]. Other polar plant extracts have also been found to be effective in inhibiting this enzyme. On the other hand, despite some authors reporting a greater inhibitory potential of isolated substances compared to complex phytoextracts [67], in our case, the activity is very similar.

## 4. Conclusions

The high phenolic content of the methanolic extract of *V. vinifera* leaves, mainly flavonoids and stilbenes, justifies its high capacity to scavenge free radicals. Especially noteworthy is its ability to capture superoxide, hydroxyl, and N free radicals, which are involved in the initiation and development of numerous diseases and pathologies, including inflammation. In addition, the extract can produce a moderate increase in intracellular ROS levels in HepG2 cells under normal growth conditions, which provides the cells greater protection against oxidative stress. This is demonstrated by the extract’s effectiveness in reducing ROS levels in cells subjected to oxidative stress. Moreover, grapevine leaves reduce the production of NO in LPS-stimulated RAW 264.7 cells even at the lowest tested concentrations and can inhibit lipoxygenase with a noteworthy IC_50_ (1.63 µg/mL). For all these reasons, grapevine leaves, an organ of the plant considered a bio-residue, can be a powerful functional food for treating diseases associated with inflammation and oxidative stress.

## Figures and Tables

**Figure 1 antioxidants-14-00279-f001:**
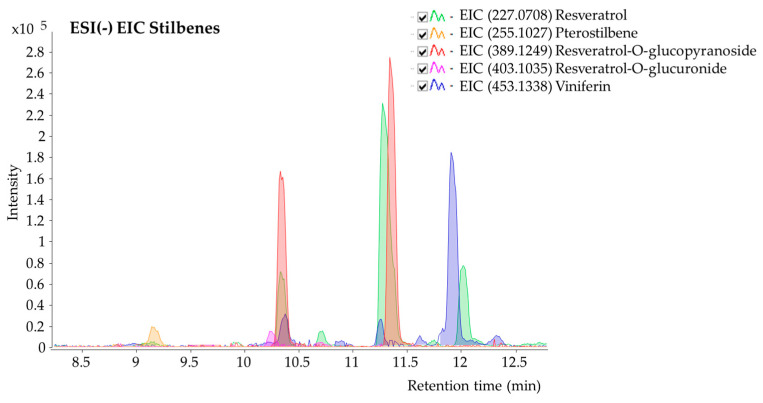
Representative Base Peak Chromatogram (BPC) in negative ion mode ESI(−) obtained from LC/MS analysis of the methanolic extract of grapevine leaves.

**Figure 2 antioxidants-14-00279-f002:**
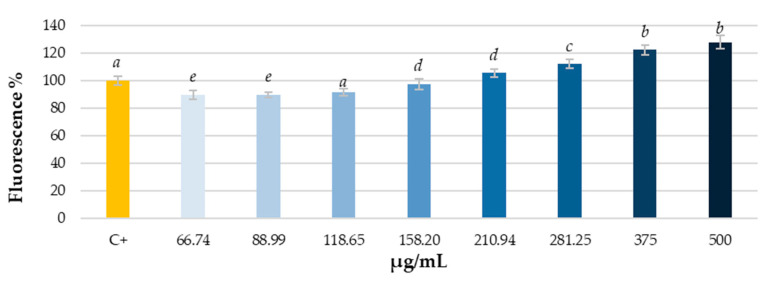
Direct effect on ROS levels in HepG2 cell culture after 90 min of treatment with different concentrations of *V. vinifera* leaf extract. Different letters indicate significant differences (ANOVA, Bonferroni *p* < 0.05).

**Figure 3 antioxidants-14-00279-f003:**
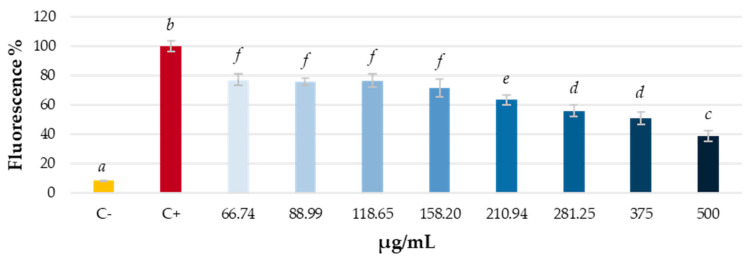
Protective effect of *V. vinifera* leaves extract on intracellular ROS levels in HepG2 cell culture after 90 min of inducing oxidative stress with H_2_O_2_ 200 mM. C−: not stressed control cells; C+: control cells under induced oxidative stress (H_2_O_2_ 200 mM). Different letters indicate significant differences (ANOVA, Bonferroni *p* < 0.05).

**Figure 4 antioxidants-14-00279-f004:**
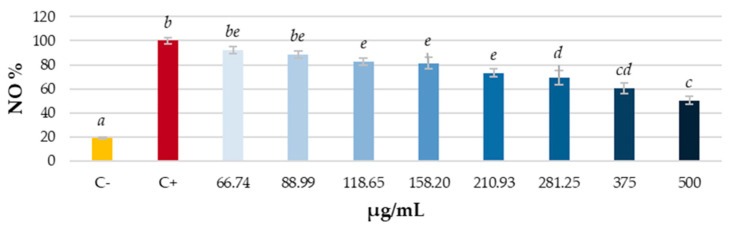
Percentage of NO produced with respect to the positive control (cells stimulated with LPS, red) in samples treated with different concentrations of grapevine leaf (blue). C−: Control cells that were not stimulated with LPS (yellow). Different letters indicate significant differences (ANOVA, Bonferroni *p* < 0.05).

**Figure 5 antioxidants-14-00279-f005:**
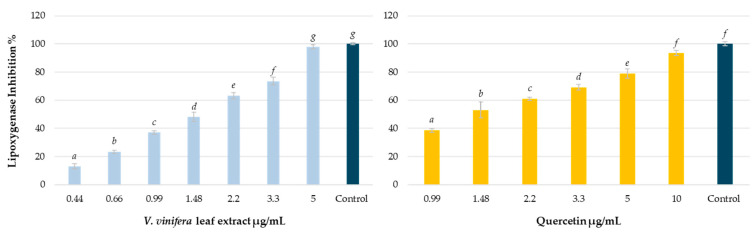
Lipoxygenase inhibitory capacity of *V. vinifera* leaf extract and quercetin. Percentage of inhibition against control. Different letters indicate significant differences (ANOVA, Bonferroni *p* < 0.01).

**Table 1 antioxidants-14-00279-t001:** Total phenol and flavonoid content of the methanolic extract of *V. vinifera* leaves. The results are expressed as mean ± S.E. of n = 3.

	Total Phenols Galic Acid mg/100 mg of the Extract	Total Flavonoids Epicatechin mg/100 mg of the Extract
*Vitis vinífera* leaves methanolic extract	25.03 ± 0.51	7.05 ± 0.07

**Table 2 antioxidants-14-00279-t002:** Phytochemical phenolic profile of the *V. vinifera* leaves extract using UHLC-QTOF-MS. Identification by standards.

Compound Group	Name	Chemical Formula	RT	Neutral Mass
Coumarin	Aesculetin	C_9_H_6_O_4_	9.1	178.0283
Coumarin	Umbelliferone	C_9_H_6_O3	10.4	162.0316
Dihydrochalcone	Phloretin	C_15_H_14_O_5_	12.7	274.0841
Dihydrochalcone	Phlorizin	C_21_H_24_O_10_	11.3	436.1369
Flavan-3-ol	(−)-Epicatechin	C_15_H_14_O_6_	9.5	290.0767
Flavan-3-ol	(+)-Catechin	C_15_H_14_O_6_	8.5	290.0740
Flavanone	Hesperetin	C_16_H_14_O_6_	12.9	302.0799
Flavone	Chrysin (Aglycone)	C_15_H_10_O_4_	15.1	254.0544
Flavone	Luteolin (Aglycone)	C_15_H_10_O_6_	12.7	286.0485
Flavone	Luteolin 7-Glucoside	C_21_H_20_O_11_	11.0	448.1004
Flavonol	Kaempferol (Aglycone)	C_15_H_10_O_6_	13.3	286.0469
Flavonol	Kaempferol 3-*O*-Glucoside	C_21_H_20_O_11_	11.5	448.1004
Flavonol	Morin (Aglycone)	C_15_H_10_O_7_	11.7	302.0483
Flavonol	Nicotiflorin (Kaempferol-*O*-Rutinoside)	C_27_H_30_O_15_	11.5	594.1637
Flavonol	Quercetin (Aglycone)	C_15_H_10_O_7_	12.0	302.0426
Flavonol	Quercetin-3-*O*-Galactoside	C_21_H_20_O_12_	11.0	464.0954
Flavonol	Quercetin-3-*O*-Glucoside	C_21_H_20_O_12_	11.0	464.0954
Flavonol	Quercetin-3-*O*-Rhaminoside	C_21_H_20_O_11_	9.2	448.1005
Flavonol	Rutin (Quercetin-3-*O*-Rutinoside)	C_27_H_30_O_16_	11.0	610.1534
Hydroxybenzoic acid	4-Hydroxybenzoic acid	C_7_H_6_O_3_	11.5	138.0347
Hydroxybenzoic acid	Gallic acid	C_7_H_6_O_5_	3.0	170.0196
Hydroxybenzoic acid	Gentisic acid	C_7_H_6_O_5_	8.3	170.0215
Hydroxybenzoic acid	Protocatechuic acid	C_7_H_6_O_4_	6.4	154.0266
Hydroxybenzoic acid	*p*-Salicylic acid	C_7_H_6_O_3_	8.3	138.0317
Hydroxybenzoic acid	Vanillic acid	C_8_H_8_O_4_	9.2	168.0423
Hydroxycinnamic acid	Caffeic acid	C_9_H_8_O_4_	9.3	180.0419
Hydroxycinnamic acid	Chlorogenic acid	C_16_H_18_O_9_	8.9	354.0920
Hydroxycinnamic acid	*p*-Coumaric acid	C_9_H_8_O_3_	10.4	164.0473
Simple phenol	Catechol	C_6_H_6_O_2_	6.8	110.0376
Simple phenol	Hydroquinone (Arbutin)	C_6_H_6_O_2_	10.5	110.0366
	Shikimic acid	C_7_H_10_O_5_	1.1	174.0528

**Table 3 antioxidants-14-00279-t003:** Tentative identification of stilbenes by LC-ESI-Q-TOF-MS in methanol extract of *V. vinifera* leaves.

Nº	Tentative Annotation	t_R_ (min)	Molecular Formula	Monoisotopic Mass	[M-H]^−^_exp_ ESI(−)	In-Source Fragments
	Stilbenes					
1	Pterostilbene	9.14	C_16_H_16_O_3_	256.1099	255.1012	[M-3CH_3_-H]^−^ = 225.0403
2	Resveratrol-*O*-glucocuronide	10.21	C_20_H_20_O_9_	404.1107	403.1616	[M-176-H]^−^ = 227.0715
3	Resveratrol-*O*-glucopyranoside	10.31	C_20_H_22_O_8_	390.1315	389.1250	[M-Hex-H]^−^ = 227.0712
4	Viniferin isomer	11.22	C_28_H_22_O_6_	454.1416	453.1354	--
5	*trans*-Resveratrol	11.24	C_14_H_12_O_3_	228.0786	227.0717	--
6	Resveratrol-*O*-glucopyranoside	11.31	C_20_H_22_O_8_	390.1315	389.1250	[M-162-H]^−^ = 227.0712
7	Viniferin isomer	11.87	C_28_H_22_O_6_	454.1416	453.1354	--
8	*cis*-Resveratrol	11.99	C_14_H_12_O_3_	228.0786	227.0716	--

**Table 4 antioxidants-14-00279-t004:** Scavenging capacity (IC_50_ (μg/mL)) of grapevine leaves methanolic extract and reference substances (ascorbic and gallic acid). Results are means of three replicates ± S.E.

IC_50_ (µg/mL)	DPPH	ABTS	O_2_^−^	OH^•^	NO^•^
Grapevine leaves extract	202.42 ± 0.01	10.59 ± 0.15	19.33 ± 0.35	155.77 ± 2.49	5.12 ± 0.19
Ascorbic acid (Gallic acid for NO^•^)	22.24 ± 0.83	5.99 ± 0.11	15.41 ± 0.13	279.53 ± 0.91	10.03 ± 0.15

## Data Availability

The raw data supporting the conclusions of this article will be made available by the authors on request.

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
