# Peer review of "Vitis vinifera L. Leaves as a Source of Phenolic Compounds with Anti-Inflammatory and Antioxidant Potential"

_antioxidants, 2025, doi:10.3390/antiox14030279_

Round 1

Reviewer 1 Report

The manuscript investigates the antioxidant and anti-inflammatory properties of Vitis vinifera L. leaves, which are commonly discarded as bio-residues in viticultural processes. The study employs UHPLC-ESI-QTOF-MS for phytochemical profiling and evaluates the extract's antioxidant potential through in vitro assays and cell culture experiments. The findings highlight the leaf extract's significant ability to scavenge free radicals, reduce oxidative stress in HepG2 cells, and inhibit lipoxygenase and NO production in LPS-stimulated RAW 264.7 cells. The results suggest that V. vinifera leaves have potential applications in functional foods and sustainable agriculture. The manuscript is well-written, methodologically sound, and presents novel findings on the bioactive properties of Vitis vinifera leaves. The study is relevant to the field of natural antioxidants and plant-based functional foods. However, some areas require clarification and revision to improve the manuscript’s impact. The authors employ rigorous phytochemical analysis and in vitro assays to validate their findings.These latter support the valorization of grapevine leaves in nutraceutical and food industries, aligning with sustainable agricultural practices. The manuscript can be accepted for publicationpending the following minor revisions

a) While multiple antioxidant assays were performed, a comparative discussion on the efficacy of V. vinifera leaf extract relative to other plant-based antioxidants would enhance the impact of the findings

b) The abstract is well-structured but could briefly mention the most potent phenolic compounds identified.

c) Minor grammatical errors and awkward phrasing in some sections should be revised. For example, "The enhancement of these by-products can improve food systems and helps the development of a sustainable agriculture" should be reworded for clarity.

d) Some references appear outdated. Consider including more recent studies on plant polyphenols and their biological activities.

The manuscript investigates the antioxidant and anti-inflammatory properties of Vitis vinifera L. leaves, which are commonly discarded as bio-residues in viticultural processes. The study employs UHPLC-ESI-QTOF-MS for phytochemical profiling and evaluates the extract's antioxidant potential through in vitro assays and cell culture experiments. The findings highlight the leaf extract's significant ability to scavenge free radicals, reduce oxidative stress in HepG2 cells, and inhibit lipoxygenase and NO production in LPS-stimulated RAW 264.7 cells. The results suggest that V. vinifera leaves have potential applications in functional foods and sustainable agriculture. The manuscript is well-written, methodologically sound, and presents novel findings on the bioactive properties of Vitis vinifera leaves. The study is relevant to the field of natural antioxidants and plant-based functional foods. However, some areas require clarification and revision to improve the manuscript’s impact. The authors employ rigorous phytochemical analysis and in vitro assays to validate their findings.These latter support the valorization of grapevine leaves in nutraceutical and food industries, aligning with sustainable agricultural practices. The manuscript can be accepted for publicationpending the following minor revisions

a) While multiple antioxidant assays were performed, a comparative discussion on the efficacy of V. vinifera leaf extract relative to other plant-based antioxidants would enhance the impact of the findings

b) The abstract is well-structured but could briefly mention the most potent phenolic compounds identified.

c) Minor grammatical errors and awkward phrasing in some sections should be revised. For example, "The enhancement of these by-products can improve food systems and helps the development of a sustainable agriculture" should be reworded for clarity.

d) Some references appear outdated. Consider including more recent studies on plant polyphenols and their biological activities.

Author Response

We warmly thank the reviewer for the time spent in revising our paper. We have addressed his/her comments and hope that the revised version will be satisfactory. The Reviewer's comments are reproduced below, with our responses following each comment.

The manuscript investigates the antioxidant and anti-inflammatory properties of Vitis vinifera L. leaves, which are commonly discarded as bio-residues in viticultural processes. The study employs UHPLC-ESI-QTOF-MS for phytochemical profiling and evaluates the extract's antioxidant potential through in vitro assays and cell culture experiments. The findings highlight the leaf extract's significant ability to scavenge free radicals, reduce oxidative stress in HepG2 cells, and inhibit lipoxygenase and NO production in LPS-stimulated RAW 264.7 cells. The results suggest that V. vinifera leaves have potential applications in functional foods and sustainable agriculture. The manuscript is well-written, methodologically sound, and presents novel findings on the bioactive properties of Vitis vinifera leaves. The study is relevant to the field of natural antioxidants and plant-based functional foods. However, some areas require clarification and revision to improve the manuscript’s impact. The authors employ rigorous phytochemical analysis and in vitro assays to validate their findings.These latter support the valorization of grapevine leaves in nutraceutical and food industries, aligning with sustainable agricultural practices. The manuscript can be accepted for publicationpending the following minor revisions

  1. While multiple antioxidant assays were performed, a comparative discussion on the efficacy of  viniferaleaf extract relative to other plant-based antioxidants would enhance the impact of the findings.

New information has been added in the discussion section comparing our antioxidant results with other parts of V. vinifera, such as grapes or byproducts from the wine industry. It also includes information on a comparison of the anti-oxidizing effect of wine and grape juice and other plant-based beverages

Other byproducts of viticulture production, such as grape pomace, have demonstrat-ed antioxidant capacity by scavenging superoxide, ABTS, and NO radicals with IC50 val-ues of 74.17, 7.79, and 15.34 µg/mL, respectively. These results lead Tapia et al. (Tapia et al., 2024) to conclude that grape pomace has potential applications in pharmacology, cosmetics, and the food industry. The results obtained with the methanolic extract of V. vinifera leaves show a greater or similar capacity to scavenge these radicals compared to grape pomace, which justifies the need for further studies to validate their practical utili-zation.

Grapes contain high levels of phenolic compounds, which confer significant antioxi-dant activity, and, consequently, health-promoting properties (Insanu et al., 2021). A study reveals that, among a range of popular beverages, both grape juice and red wine are among the top four with the highest in vitro antioxidant potential of all the beverages evaluated, surpassing orange juice, iced tea, and apple juice, although falling behind pomegranate juice (Seeram et al., 2008). Grapes exhibit IC50 values for DPPH (270 ± 1 µg/mL) and ABTS (40 ± 3 µg/mL) that are notably higher than those reported in this study, suggesting that the leaves possess superior antioxidant capacity. Furthermore, while the skin and seeds of these fruits are recognized as antioxidant-rich sources with potential for use as dietary supplements, their antioxidant capacity is lower than that of the leaves in these assays, with the exception of the seeds in DPPH capture (IC50: 30.6 ± 3.4 µg/mL (Castro-López et al., 2019).

  1. The abstract is well-structured but could briefly mention the most potent phenolic compounds identified.

A sentence in this respect has been included in the abstract.

Hydroxybenzoic acids, flavonols and stilbenes are the main phenolic compounds identified.

c) Minor grammatical errors and awkward phrasing in some sections should be revised. For example, "The enhancement of these by-products can improve food systems and helps the development of a sustainable agriculture" should be reworded for clarity.

The text has been reviewed and the sentence reworded, we hope is clearer now:

The enhancement of these by-products can be highly beneficial to food systems and con-tribute to the development of sustainable agriculture.

d) Some references appear outdated. Consider including more recent studies on plant polyphenols and their biological activities.

Bibliography has been updated.

Reviewer 2 Report

Dear Authors,

This is an interesting study on Vitis vinifera leaves and their antioxidant and anti-inflammatory properties. Your research shows the potential of grapevine leaves as a source of bioactive compounds. It also highlights the value of viticultural by-products for sustainable agriculture and functional foods.

I have no major comments.

Kind regards.

Here are specific comments:

The abstract: at the end of the sentence dot is missing

Materials and Methods:

quercetin-3-O-glucoside O should be in italic

help to preserve the anthocyanins [14,15] (space is missing)

5 μL of five different concentrations of the extract were added to each of a 96 plate and mixed with 80 μL of 10% Folin-Ciocalteau.

Add details about Gallic acid and Epicatechin standard curve preparation in each method described

Through manuscript cyanidin-3-glycoside or cyaniding-3-O-glucoside, etc…

The separation was performed in a reverse-phase column Zorbax Eclipse XDB-C18 (4.6 × 50 mm, 1.8 μ m) (Agilent Technologies, Walb…Germany - the details are missing)

4 h of incubation, the wells content was aspirated and 100 μL of DMSO were added to dissolve formazan crystals.

post hoc comparison (p<0.05) – is this p0.05

the red-ox balance, should be redox balance

Author Response

We warmly thank the reviewer for the time spent in revising our paper. We have addressed his/her comments and hope that the revised version will be satisfactory. The Reviewer's comments are reproduced below, with our responses following each comment in light blue.

The abstract: at the end of the sentence dot is missing.

The dot has been added.

Materials and Methods:

quercetin-3-O-glucoside O should be in italic.

Corrected.

help to preserve the anthocyanins [14,15] (space is missing)

A space has been added.

5 μL of five different concentrations of the extract were added to each of a 96 plate and mixed with 80 μL of 10% Folin-Ciocalteau.

Corrected.

Add details about Gallic acid and Epicatechin standard curve preparation in each method described

Details of used concentrations have been given

Through manuscript cyanidin-3-glycoside or cyaniding-3-O-glucoside, etc…

Corrected.

The separation was performed in a reverse-phase column Zorbax Eclipse XDB-C18 (4.6 × 50 mm, 1.8 μ m) (Agilent Technologies, Walb…Germany - the details are missing).

The information has been added.

4 h of incubation, the wells content was aspirated and 100 μL of DMSO were added to dissolve formazan crystals.

post hoc comparison (p<0.05) – is this p ≤ 0.05

Corrected.

the red-ox balance, should be redox balance

Corrected.
